# A Novel Symbiotic Formulation Reduces Obesity and Concomitant Metabolic Syndrome in Rats by Raising the Relative Abundance of Blautia

**DOI:** 10.3390/nu15040956

**Published:** 2023-02-14

**Authors:** Xiu-Rong Wu, Zhen-Zhen Chen, Xi-Lan Dong, Qiu-Ping Zhao, Jun Cai

**Affiliations:** 1Hypertension Center, People’s Hospital of Zhengzhou University, Zhengzhou 450001, China; 2Hypertension Center, Fuwai Central China Cardiovascular Hospital, Zhengzhou 451464, China; 3Hypertension Center, Fuwai Hospital State Key Laboratory of Cardiovascular Disease, National Center for Cardiovascular Diseases Chinese, Academy of Medical Sciences and Peking Union Medical College Beijing, Beijing 102308, China

**Keywords:** overweight, fat accumulation, probiotics, prebiotics, intestinal flora, metabolic disorder

## Abstract

Obesity is regarded as an abnormal or excessive buildup of fat that may be bad for health and is influenced by a combination of intestinal flora, genetic background, physical activity level and environment. Symbiotic supplementation may be a realistic and easy therapy for the reversal of obesity and associated metabolic problems. In this study, we chose two Bifidobacterium species, three Lactobacilli species and four prebiotics to make a new symbiotic formulation. High or low doses of the symbiotic were administered to rats, and biochemical indicators were recorded to assess the biological effects in a high-fat-diet-induced rat model. The underlying mechanisms were explored by integrating 16S rRNA sequencing with an extensively targeted metabolome. High-dose symbiotic supplementation was effective in reducing obesity and concomitant metabolic syndrome. The high-dose symbiotic also significantly increased the abundance of Blautia, which was negatively correlated with taurocholic acid and the main differential metabolites involved in amino acid and bile acid metabolism. While the low-dose symbiotic had some therapeutic effects, they were not as strong as those at the high dose, demonstrating that the effects were dose-dependent. Overall, our novel symbiotic combination improved plasma glucose and lipid levels, shrunk adipocyte size, restored liver function, increased the abundance of Blautia and adjusted bile acid and amino acid metabolism.

## 1. Introduction

The World Health Organization (WHO) classifies adults with a body mass index (BMI) of 30 to be obese, and as of 2016, more than 650 million persons were obese worldwide [1]. Despite growing concern, the prevalence of obesity is rising globally [2]. Obesity contributes to the progression of cardiovascular disease and death [3]. Furthermore, some research indicates that overweight and obese teenagers are more prone to being bullied compared with their normal-weight peers [4]. Excessive fat deposition in obesity has a complex origin but is largely thought to be the outcome of an energy intake/expenditure imbalance [5]. Food provides people with more energy than they need for daily consumption. Traditional treatments for weight loss, such as lifestyle changes, medication and bariatric surgery, are either difficult to follow or have substantial adverse effects [2]. As a result, there is an increasing need to identify long-term and efficacious methods of losing weight.

The adult gut contains trillions of microorganisms [6]. Obesity is frequently associated with an imbalance of gut flora [7]. Probiotics are defined by the WHO as live microorganisms that are helpful to the body, and they are increasingly being shown to treat obesity and its underlying metabolic syndrome. Obese and diabetic mice showed decreased body weight and improved glucose tolerance when fed with *Bifidobacterium animalis ssp. lactis* 420 (B420) [8]. *Bifidobacterium breve* strain B-3 (B-3) modulates fat metabolism and prevents weight gain [9]. Additionally, some lactobacilli have also been reported to improve blood lipid levels, suppress liver lipid accumulation and restore liver function [10,11]. Prebiotics, such as inulin, oligofructose and pectin, are nonviable dietary components that provide health advantages to the host. Inulin is a colorless, neutral-tasting high-fermentable fiber that is an FDA-approved prebiotic [12]. Recently, it has been shown to enhance the digestive system, regulate the lipid metabolism and modulate insulinemia and blood glucose levels [13]. Oligofructose is one of the top three internationally recognized prebiotics. It improved liver steatosis relative to placebo and enhanced Bifidobacterium [14]. The addition of oligofructose to the diet can inhibit lipid absorption inside the gut, resulting in reductions in body weight and body fat [15]. Pectin can impede starch digestion, resulting in decreased glucose absorption and blood glucose levels by blocking alpha-amylase activity in the blood, as well as reduce body weight and improve blood lipids [16,17]. The traditional Mediterranean diet is distinguished by the consumption of polyphenol-rich foods and is highly recommended by nutritionists as a healthful eating pattern. Phenolic compounds have been shown to regulate intestinal flora to protect intestinal health [18,19]. Cranberry powder is frequently used as a functional supplement because of its high polyphenol content, and it has been found to be beneficial in reducing hepatic lipid droplet formation and blood triglyceride in a mice model induced by a high-fat diet (HFD) [20]. Therefore, polyphenol-rich cranberry powder may assist to regulate their blood triglyceride and hepatic lipid droplet formation. Overall, bifidobacteria, lactobacilli and certain prebiotics have considerable health impacts on fat formation, blood glucose and lipid regulation, and liver function restoration.

As probiotic research has advanced, increasing numbers of uncontrolled probiotic combinations known as symbiotics have appeared in the consumer marketplace. In response, the International Scientific Association of Probiotics and Prebiotics defined a symbiotic as a combination of living microorganisms and substrate(s) selectively used by host microorganisms that benefits the host’s health [21]. The prebiotics in these formulations are designed to help the selected probiotics flourish. There is abundant evidence that supplementation with a symbiotic is more beneficial than that with a single probiotic strain for preventing or controlling excess weight and obesity. According to one study, a supplementary mix of three probiotic strains enriched *Akkermansia mucinphila*, negatively affecting nonesterified fatty acid (NEFA) and energy metabolism in rats [22]. Additionally, a study of allulose and two probiotic bacterial strains in mice demonstrated that the symbiotic combination was more efficient than either the probiotics or allulose alone in suppressing obesity caused by diet and concomitant metabolic syndrome by lipid metabolism modulation [23]. Although the probiotics and prebiotics mentioned above have considerable effects on obesity when given alone, the therapeutic effects of combined supplementation are uncertain.

This study examined how a new symbiotic affected obesity symptoms in HFD-induced rats. The symbiotic was formulated with the following components: two strains of *Bifidobacterium* that have been shown by research to help people lose weight (B420 and B-3); three strains of *Lactobacillus* that have strong potential to decrease blood levels of cholesterol and glucose (*Lactobacillus acidophilus* strain JYLA-191, *L. rhamnosus* strain JYLR-005 and *L. plantarum* strain JYLP-002); and four prebiotics that modulate glycolipid metabolism (apple pectin, inulin from Jerusalem artichokes, fructo-oligosaccharides and cranberry powder). The prebiotics in our formulation are intended to assist chosen probiotics to grow and execute their physiological function more effectively, making the benefits of probiotics more powerful and long-lasting. High-dose supplementation of this symbiotic alleviated weight gain, improved obesity-related metabolic syndrome and altered the structure and function of the gut flora, with particularly strong enhancement of *Blautia*, the potential probiotics associated with obesity. Our novel symbiotic formula may be effective for increasing the abundance of *Blautia*, while preventing and/or regulating the course of obesity and concomitant metabolic syndrome.

## 2. Materials and Methods

### 2.1. Animals and Diets

Male Sprague Dawley rats (age, 5 weeks) were acquired from the Vital River Laboratory (Beijing, China) and raised in the Experimental Animal Center of Fuwai Hospital. The rats were housed (2–3 rats per cage) and maintained under controlled temperature (20–25 °C) and 12 h light/dark conditions. Except for the normal diet (ND) control group, which was fed a chow from HFK Bio-Technology Co., Ltd. (1025, Beijing, China), all experimental animals were fed a HFD containing 45% fat from Medicience Ltd. (cat. MD12032, Yangzhou, China). Throughout the trial, the rats had full access to water and food. The rats were measured for body length and then euthanized at the end of the experiment. Fat, liver and plasma samples were obtained and kept in a freezer at −80 °C until examination. All animal protocols were in compliance with all applicable ethical requirements and were approved by the Institutional Animal Care and Use Committee of the Experimental Animal Center, Fuwai Hospital, National Center for Cardiovascular Diseases, China.

### 2.2. Symbiotic Treatments and Groups

Our innovative symbiotic mix was prepared using powdered formulations of five probiotic strains (*B. animalis* ssp. *lactis* 420, *B. breve* B-3, *L. acidophilus* JYLA-191, *L. rhamnosus* JYLR-005 and *L. plantarum* JYLP-002) mixed with four prebiotics (apple pectin, inulin, fructo-oligosaccharides and cranberry powder), all of which were provided by HaoDingRui Biotech Co., Ltd. (Beijing, China). The exact formula of the symbiotic is listed in Appendix A. This symbiotic mixture was dissolved in 2 mL of phosphate-buffered saline (PBS) and fed to each rat in HFD_L or HFD_H for 10 weeks. Following 10 weeks of HFD-induced obesity, rats were split into three groups: HFD with PBS control group (HFD_C, n = 8), HFD with low-dose (10^8^ colony-forming units (CFU)) symbiotic (HFD_L, n = 7) and HFD with high-dose (10^9^ CFU) symbiotic (HFD_H, n = 7). The ND group received the same quantity of PBS.

### 2.3. Body Weight and Biomarkers

Over the 20-week period, we recorded body weight weekly. An EchoMRI instrument (QMR06-090H, Niumag Analytical Instrument Corporation, Suzhou, China) was used to determine lean and fat body mass. Commercial kits (BioSino Bio-Technology and Science Inc., Beijing, China) were used to measure plasma triglycerides (TG) and total cholesterol (TC). Aspartate aminotransferase (AST) and alanine aminotransferase (ALT) were examined using commercial kits from Njjcbio (Nanjing, China). The Roche blood glucose meter and Roche test papers (Basel, Switzerland) were used to measure fasting blood glucose every 2 weeks following a 6–8 h fasting period. Glucose tolerance was determined by fasting rats for 6–8 h before weighing them and administering glucose intraperitoneally (2 g/kg body weight).

### 2.4. Histopathological Analysis

Liver and fat samples from each rat were removed, fixed in 4% paraformaldehyde and embedded in paraffin in preparation for staining with hematoxylin and eosin (H&E) using a Varistain Gemini slide stainer (Thermo Scientific, Waltham, MA, USA). Liver samples from the rats were also embedded in optimal cutting temperature compound (OCT, Solarbio, Beijing, China) and stored at −80 °C before staining with an oil red O staining kit (Solarbio, Beijing, China). The kit was used in accordance with the manufacturer’s instructions for drying and staining frozen tissue. Images were obtained by a slicing scanner at 20× magnification (Pannoramic SCAN, 3D HISTECH, Budapest, Hungary).

### 2.5. The 16S rRNA Sequencing of Fecal Samples and Bioinformatics Analysis

Total genomic DNA was isolated from fecal samples using an OMEGA Soil DNA Kit (cat. M5635-02, Omega Bio-Tek, Norcross, GA, USA) and kept at −20 °C before further analysis. The V3-V4 region of bacterial 16S rRNA genes was amplified with primers 338F (5′-ACTCCTACGGGAGGCAGCA-3′) and 806R (5′-GGACTACHVGGGTWTCTAAT-3′). After the individual quantification phase, amplicons were pooled in equal amounts, and paired-end 2 × 250 bp sequencing was performed on an Illumina NovaSeq platform using a NovaSeq 6000 SP Reagent Kit (500 cycles) at Shanghai Personal Biotechnology Co., Ltd. (Shanghai, China). QIIME2 2019.4 was used for microbiome bioinformatics with slight modifications to the standard tutorials (https://docs.qiime2.org/2019.4/tutorials/) [24].

Sequence data analyses were mainly performed using the QIIME2 and R packages (v. 3.2.0). Based on the ASV table in QIIME2, alpha diversity was estimated using the Shannon diversity index, Chao1 richness estimator, Good’s coverage index and Pielou’s evenness index and illustrated by box plots. The structural variance of microbial communities between samples was also investigated using Bray–Curtis metrics [25] and displayed using principal coordinate analysis (PCoA). With default parameters, we performed a linear discriminant analysis effect size (LEfSe) to determine whether there were different levels of abundancy among taxa across groups [26]. Phylogenetic investigation of communities by reconstruction of unobserved states (PICRUSt2, Version 2.2.0) predicted microbial functions using two databases: Metabolic Pathways from All Domains of Life (MetaCyc; https://metacyc.org/) and Kyoto Encyclopedia of Genes and Genomes (KEGG; https://www.kegg.jp/). 

### 2.6. Broadly Targeted Liquid Chromatography-Tandem Mass Spectrometry (LC-MS/MS) Measurement of Metabolites in Plasma Samples

Metware Biotechnology Co., Ltd. (Wuhan, China) quantified fecal metabolites. Samples were thawed on ice and vortexed (10 s) after removal from −80 °C. Each sample (50 μL) was diluted with 300 μL of extraction solution (acetonitrile: methanol = 1:4, *v*/*v*) containing internal standards and transferred to a 2 mL microcentrifuge tube. Samples were vortexed (3 min) and then centrifuged at 12,000 rpm (10 min, 4 °C). A 200 μL aliquot of supernatant was collected, placed in (30 min, −20 °C) and then centrifuged at 12,000 rpm (10 min, 4 °C). Subsequently, a 180 μL aliquot of supernatant was subjected to LC-MS analysis using an LC-electrospray ionization (ESI)-MS/MS system (ultrapure LC (UPLC), ExionLC AD, https://sciex.com.cn/ accessed on 11 May 2022; MS, QTRAP^®^ System, https://sciex.com/ accessed on 11 May 2022). The analytical conditions were as follows, UPLC HSS T3 C18 (1.8 µm, 2.1 mm × 100 mm) column (Waters ACQUITY, Massachusetts, USA); column temperature, 40 °C; flow rate, 0.4 mL/min; injection volume, 2 μL; solvent system, water (0.1% formic acid): acetonitrile (0.1% formic acid); and gradient program, 95:5 *v*/*v* at 0 min, 10:90 *v*/*v* at 11.0 min, 10:90 *v*/*v* at 12.0 min, 95:5 *v*/*v* at 12.1 min and 95:5 *v*/*v* at 14.0 min. A QTRAP mass spectrometer, operated in positive and negative ion mode and controlled by Analyst 1.6.3 software (https://sciex.com/ accessed on 11 May 2022), was used to collect linear ion trap (LIT) and triple quadrupole (QQQ) scans. The ESI parameters were as follows: temperature, 500 °C; voltage, 5500 V (+) and -4500 V (−); gas I, 55 psi; gas II, 60 psi; and curtain gas 25 psi. For the metabolites eluted during each phase, a specific set of several reaction-monitoring transitions was monitored.

OPLS-DA was analyzed using the MetaboAnalystR package in R software (Version 1.0.1). Heatmap was performed and displayed by the ComplexHeatmapR package in R software (Version 2.8.0). 

### 2.7. Formatting of Mathematical Components

Lee index is calculated using the following formula.
$$\mathrm{Lee}\;\mathrm{index}=\mathrm{weight}{\left(g\right)}^{\frac{1}{3}}*10/\mathrm{body}\;\mathrm{length}(\mathrm{mm})\;$$

### 2.8. Statistical Analysis

Data are presented as mean ±SEM. An unpaired Student’s *t*-test was used to compare the ND and HFD C groups. Data from three or more groups were compared using one-way analysis of variance (ANOVA). Two-way ANOVA was used for comparisons involving two components. Two-way mixed-effects ANOVA was used to examine the results of repeated measurements from the animals. GraphPad Prism v. 8.0.2 was used for all statistical calculations. A *p* value < 0.05 was considered statistically significant. 

## 3. Results

### 3.1. Symbiotic Treatment Reduced Body Weight and Body Fat in Obese Rats

We created an HFD-induced obesity model in rats to investigate the effect of the symbiotic on body weight and fat. All rats had equal body weights before the administration of the HFD at week 0. Body weight increase symptomatic of obesity occurred 3 weeks after initiation of the HFD in all three HFD feeding groups. While body weight continued to increase until the end of the experiment (20 weeks) in the HFD control group (HFD_C), body weight in the high-dose symbiotic group (HFD_H) tended to be lower (5.8%) than that in the HFD_C group at week 20 (*p* = 0.068; Figure 1A). In comparison to the ND control group, HFD_C rats had significantly higher Lee index (4.2%) and fat mass percentages (46.7%) but lower lean mass percentages (28.7%) (Figure 1B,C). By contrast, the Lee index (3.6%) and fat mass percentage (17.8%) were significantly lower in the HFD_H group than in the HFD_C group at week 20 (*p* < 0.05 and *p* < 0.001, respectively; Figure 1B,C). By week 20, the low-dose symbiotic group (HFD_L) also had a lower fat mass percentage (15.4%) than the HFD_C group (*p* < 0.05; Figure 1C). However, the lean mass percentage values in both the high- and low-dose symbiotic groups were not significantly different from that in the HFD_C group at week 20 (Figure 1D). Provision of the HFD generated epididymal adipocyte hypertrophy (177.6%) (Figure 1E, HFD_C vs. ND), but the high-dose symbiotic significantly reversed the increase in adipocyte sizes (56.9%) in HFD_H rats at week 20 (*p* < 0.001, HFD_H vs. HFD_C). Representative photos of rats at week 20 in each group are shown in Figure 1F. These findings demonstrated that supplementation with our novel symbiotic appeared to reduce the increase in body fat caused by HFD feeding.

### 3.2. Symbiotic Treatment Improved Dysglycemia and Dyslipidemia in Obese Rats

Obesity is associated with poor blood glucose control. As shown in Figure 2A, HFD feeding increased (20.0%) the fasting blood glucose level in the HFD_C groups (*p* < 0.0001, vs. ND) at week 20, which was significantly reversed (13.0%) at week 20 by high-dose symbiotic supplementation in HFD_H (*p* < 0.01, vs. HFD_C). We also observed a downward trend (9.0%) in HFD_L compared with HFD_C, but the difference was not significant (*p* = 0.06). Glycemia rose significantly in all four groups after an intraperitoneal glucose tolerance test, peaking at 30 min post challenge. Compared with the ND group, HFD rats had serious impairments in glucose tolerance, which was remedied by high-dose symbiotic intervention in HFD_H (Figure 2B). Area under the curve values differed significantly between HFD_C and HFD_H (*p* < 0.05; Figure 2C). Dyslipidemia is a typical consequence of obesity. As shown in Figure 2D, HFD feeding increased plasma TG levels significantly (100.6%), which were effectively decreased with both high- (33.1%) and low-dose (36.7%) symbiotic supplementation (*p* < 0.05). Meanwhile, the symbiotic intervention also significantly lowered (41.1%) plasma TC (*p* < 0.05), but only in the HFD_H group (Figure 2E). These findings revealed that the beneficial effects of the symbiotic on plasma lipid and glucose levels were stronger at the high dose than at the lose dose. 

### 3.3. Symbiotic Treatment Inhibited Liver Lesions and Restored Liver Function in Obese Rats

In both animals and humans, the liver contributes greatly to metabolic balance. To assess the contribution of our prebiotic and probiotic combination on the physiological function of the liver, we evaluated biochemical indicators of liver function as well as histological alterations in liver tissue towards the conclusion of the experiment. As shown in Figure 3A,B, HFD feeding significantly increased the levels of ALT (43.1%) and AST (54.1%) in the HFD_C group (*p* < 0.05 vs. ND). Symbiotic supplementation alleviated the increase in ALT level (47.5%) in HFD_H compared with that in HFD_C. Although there were no significant differences in AST levels among the three HFD groups, both the HFD_L and HFD_H groups had lower levels than the HFD_C group. Next, we examined histopathological changes. The livers of HFD_C rats had a greater number of lipid droplet vacuoles when compared to ND rats; however, this was alleviated by the high-dose symbiotic in HFD_H rats (Figure 3C). Correspondingly, the obvious results under oil red O staining indicate that the HFD_C rats accumulated more fat in the liver compared with ND rats, but high-dose symbiotic supplementation reduced hepatic steatosis and fat buildup (Figure 3D). The difference in the proportion of oil red O staining (% area) between the HFD_H and HFD_C groups was significant (*p* < 0.001; Figure 3F). These results suggest that long-term intake of our symbiotic formulation alleviated the liver dysfunction caused by HFD feeding, including reductions in fat accumulation and liver injury.

### 3.4. Symbiotic Treatment Altered the Composition and Function of the Gut Microbiota in Obese Rats

To investigate whether the symbiotic-induced decrease in fat storage was due to alterations in the gut flora, we examined fecal samples for between-group differences in terms of total species richness and evenness (alpha diversity), and microbial composition (beta diversity). The symbiotic dosage had no effect on alpha diversity (Figure 4A), but it did impact beta diversity (Figure 4B). The ND group and HFD_H group have greater intra-group dispersion, while the HFD_C and HFD_L groups are relatively clustered. To explore the differential microbiota in all groups, we performed LEfSe analysis. Figure 4C shows the microbiota significantly enriched within each group for all taxonomic levels. Appendix A displays the phylogenetic relationships between these taxa using a cladogram. When comparing the two experimental groups, the relative abundance at the phylum level, particularly for Firmicutes and Bacteroidetes, did not show significant differences under symbiotic supplementation (Figure 4D). This might be explained by the technique’s limitations; i.e., sequencing of the 16S rRNA gene amplicon does not cover the whole gut flora. By contrast, compared with HFD_C, the symbiotic significantly elevated the abundance of *Blautia*, Gammaproteobacteria and *HTCC2188*, and showed a trend toward increasing the abundance of *Roseburia* in HFD_H (Figure 4E). These findings suggest that high-dose symbiotic treatment dramatically altered the makeup of the gut microbiota.

Functional annotations can assist with separating differences in metabolic phenotypes between microorganisms and hosts and analyzing their interactions, which are fundamentally linked with the composition of intestinal flora. For functional annotation analysis, we used the KEGG and MetaCyc databases. The following KEGG pathways were enhanced by high-dose symbiotic supplementation in obese rats: biosynthesis of other secondary metabolites, amino acid metabolism, transcription and metabolism of cofactors and vitamins, and energy metabolism. Interestingly, the relative abundance of infectious disease pathways was reversed in the HFD_H group as compared to the HFD_C group (Figure 5A). Comparison of the functional bacterial genes across the four groups in MetaCyc revealed that that the administration of high-dose symbiotic upregulated the activity of many pathways, including the following: fatty acid and lipid degradation, metabolic regular biosynthesis, secondary metabolite degradation, polymeric compound degradation, glycan degradation, L-glutamate and L-glutamine biosynthesis, and aldehyde degradation (Figure 5B). As shown in Figure 4, Figure 5 and Appendix A, there was variability in the structure of the gut flora, and in the functional modules of the flora, among the four groups of rats.

### 3.5. Impact of Symbiotic Treatment on the Metabolic Profile in Blood Plasma

Because certain products of gut microbiota fermentation can reach the bloodstream and impact host physiology, we used high-throughput LC-MS to examine the host metabolic profile in fasting plasma of rats and presented the results as a histogram (Appendix A). To discriminate between the metabolic profiles across groups, we performed clustering based on orthogonal partial least squares-discriminant analysis (OPLS-DA), which largely separated the plasma samples according to treatment group (Figure 6A). Separation of the HFD_L and HFD_H groups from HFD_C indicates that the symbiotic had a strong influence on the metabolites in rat plasma. Furthermore, the observed distinction between the HFD_L and HFD_H groups suggest that the dose also had an influence on the plasma metabolites. Similarly, according to previous studies, symbiotics highly influence the plasma metabolite profile [27]. 

Sixty-five characteristic metabolic molecules (Appendix A) were identified by the OPLS-DA model (VIP score > 1; *p* < 0.05; fold change ≥2 or ≤0.5). Most of these characteristic metabolic molecules belong to the fatty acyl, glycerophospholipids and organic acids. These molecules were then visualized using hierarchical clustering of samples, with the results displayed as both a heatmap and a tree diagram. As shown in Figure 6B and Appendix A, the metabolite levels varied greatly between different rats, with particularly large differences within HFD_C and HFD_L, and relatively concentrated in the ND and HFD_H groups. Volcano plots were used to investigate the variations in metabolite content between sets of two-group comparisons. As shown in Figure 6C, varying the fat ratio of the diet, intervention with the symbiotic, and the symbiotic dosage all affected the plasma metabolites in rats. 

To explore possible relationships among the alterations in gut flora and metabolites that emerged between the HFD_H and HFD_C rats, we performed Spearman correlation analysis on 67 core microorganisms and 46 key plasma metabolites and illustrated the results using a heatmap (Figure 6D). Some of these metabolites—particularly bile acids and their intermediary metabolites, such as alpha, beta and gamma muricholic acids, cholic acid, 7-ketolithocholic acid and 12-ketolithocholic acid—were significantly decreased in the HFD_H and were positively linked with *f__Methylocystaceae*. However, negative correlations were observed with o__Streptophyta; in particular, reduced taurocholic acid was negatively correlated with *g__Blautia*. Furthermore, the (±) 4-hydroxy-docosahexaenoic acid (HDHA) level of HFD_H was also considerably lower and was positively associated with *g__Clostridium* and *f__Methylocystaceae* but negatively correlated with *f__Alcanivoracaceae*, o__Oceanospirillales, *g__Alcanivorax*, o__HTCC2188, c__Gammaproteobacteria and *g__Blautia*. Glu-Phe levels were positively correlated with c__Gammaproteobacteria and o __HTCC218. L-arginine levels were positively correlated with c__Gammaproteobacteria and *g__Blautia* but negatively correlated with *g__Weissella*, *f__Leuconostocaceae* and *f__Aerococcaceae*. These findings indicate a link between the gut bacteria and plasma metabolites, implying that microbial abundance alterations may regulate metabolic outcomes.

## 4. Discussion

Obesity is closely associated with metabolic diseases and health problems. Many researchers have spent considerable time and money developing drugs and foods which are effective in controlling obesity. How the intestinal flora affects obesity and concomitant metabolic syndrome has only become clear in the last decade [28,29]. Although it has been shown that *Bifidobacterium*, *Lactobacillus* and certain prebiotics have different biological functions related to obesity when used individually, the potential therapeutic effects and mechanisms of action of their combination are unknown. By integrating plasma biochemical parameters, the gut microbiome and plasma metabolomics data, we demonstrated that our novel symbiotic has the ability to reduce fatty buildup and improve obesity-related metabolic disorders and liver damage by modifying the structure and function of the intestinal flora to regulate the metabolism of bile acids, amino acids and some other small molecules.

Many strains of *Lactobacillus* and *Bifidobacterium*, the key components of most probiotic formulations, have been demonstrated to have beneficial effects on obesity or associated metabolic consequences in animal models, including aberrant lipid profile and glucose level or glucose tolerance and liver fat accumulation [8,9,10,30]. Some prebiotics derived from vegetables and fruits may also help to reduce body weight and fat, improving metabolic parameters and preventing liver damage [31,32,33,34,35]. In this study, a high dose of our symbiotic showed high potential for weight control, as well as effects on glycemic control. These results are highly consistent with previous studies on *B. animalis ssp. lactis* 420, *B. breve* B-3, fructo-oligosaccharides and other prebiotics [8,9,20,31]. Conditions involving abnormally high levels of plasma lipids, such as hypertriglyceridemia and hypercholesterolemia, are associated with risk of cardiovascular disease. Our findings showed that plasma TG levels were considerably decreased with high- or low-dose symbiotic supplementation, while the plasma TC levels were significantly decreased only in the high-dose group. These effects were also mentioned in previous studies of probiotic/prebiotic supplementation in rodents or humans [9,10,32]. Plasma ALT and AST levels are regarded as markers of liver damage in a variety of diseases, including NAFLD [36]. In this study, we observed that high-dose symbiotic supplementation significantly reduced plasma ALT levels and attenuated hepatic fat accumulation, but the effect on AST was not obvious. Previous studies elucidated the hepatoprotective effects of *B. animalis *ssp. *lactis* 420, *L. acidophilus* LA5, *L. plantarum* mixtures and pectin [10,11,30,33]. The therapeutic effect of this symbiosis on the metabolic disorders caused by obesity is probably composed of three components: (1) *B. animalis *ssp. *lactis* 420, *B. breve* B-3 and inulin to regulate plasma glucose levels, (2) Lactobacillus plantarum and inulin to regulate lipid levels, and (3) *B. animalis *ssp. *lactis* 420, *L. plantarum* JYLP-002, *L. acidophilus* JYLA-191 and pectin to participate in the recovery of liver function. In conclusion, symbiotic supplements can be used to simultaneously compensate for single strain shortages and work synergistically to reverse dietary obesity and concomitant metabolic syndrome. 

*Blautia* is an anaerobic probiotic that occurs widely in the gut and feces of mammals [37]. The supplementation of feruloylated oligo- and polysaccharides (FOPS), a dietary fiber produced from maize, dramatically enhanced the abundance of *Blautia* in the feces in mice on HFD. Another study also showed that supplementation of oligofructose significantly increased the abundance of *Blautia* [38]. Compared to mice on the HFD alone, those on the F-FOPs + HFD also lost body and tissue weight, demonstrating a negative connection between the abundance of *Blautia* and indicators of obesity and concomitant metabolic syndrome [39]. *Blautia* deficiency has been identified in the intestines of obese children and has been linked to insulin resistance [40], whereas a low-fat diet increased the abundance of *Blautia* in healthy young adults [41]. In a study of Japanese people aged 20–76, *Blautia* was shown to be inversely correlated with visceral fat area [42]. In our study, supplementation with the high-dose symbiotic effectively increased the relative abundance of *Blautia*, which was inversely correlated with levels of plasma taurocholic acid, (±)4-HDHA and Glu-Phe. Similar to previous studies, the increased abundance of *Blautia* might be attributed to the action of oligofructose in our formulation. This extraordinary result encapsulates the potential application of our novel symbiotic in weight and obesity control. Although in vitro cultivation techniques for *Blautia* are still being developed, our findings may inspire novel approaches for the development of new *Blautia*-beneficial dietary or pharmaceutical supplements.

Our research found that symbiotic supplementation changed the plasma metabolite profile, including bile acids, amino acids and certain small molecules that affect host health. Bile acids are chemicals secreted by the liver and signaling molecules that can be used to coordinate the regulation of metabolism through the nuclear farnesoid X receptor (FXR) and the Takeda G protein-coupled receptor 5 (TGR5). Bile acid production and excretion are key routes of cholesterol and lipid catabolism, and are thus related to a number of metabolic disorders, such as obesity, insulin resistance and NAFLD [43]. According to several recent studies, the intestinal flora regulates bile acid metabolism [44]. Bile acids in the intestine are broken down into amino acids and free bile acids by bile salt hydrolases secreted by probiotics [45]. Some of the free bile acids are converted to secondary bile acids which then bind to the TGR5 and FXR receptors, decreasing blood glucose and lipid levels [46]. The intestinal flora not only influences secondary bile acid metabolism, but it also limits bile acid production in the liver through decreasing FXR inhibition in the ileum. A reduction in gut microorganisms elevated tauro-β-muricholic acid (MCA), a rodent-specific FXR antagonist, and decreased the diversity of the bile acid pool in germ-free or antibiotic-treated mice [47]. In another study, CYP2C-deficient mice exhibited decreased MCAs and changes in bile acid composition, and obesity and associated metabolic abnormalities did not emerge after being fed a HFD, compared to wild-type mice [48]. Similar reductions in MCAs and variations in bile composition were seen in this study, and the reduction in MCAs was related to alterations in the intestinal flora. In a randomized, double-blind, placebo-controlled clinical study conducted in Spain in a hyperlipidemic population, researchers used Lactobacillus plantarum with high BSHs activity for a 12-week intervention. The results showed that TG and TC levels were significantly lower in the probiotic intervention group than in the control group [49]. This strongly suggests that the reduction in blood glucose lipids levels we observed is most likely due to the hydrolysis of bile acids by Lactobacillus plantarum. Interestingly, symbiotic supplementation altered not only bile acid metabolism but also amino acid metabolism, for example, by increasing Glu-Phe and L-arginine. Glu-Phe is a biologically active dipeptide that may inhibit the expression of sterol regulatory element-binding protein-1c (SREBP-1c) and its adipogenic target genes by activating AMP-activated protein kinase, resulting in decreased hepatic fat deposition in mice [50]. L-arginine is a functional amino acid that may play an important role in body metabolism and physiology via the L-arginine-nitric oxide pathway, possibly lowering blood levels of free fatty acid, triacylglycerol and cholesterol in many species [51]. The effect of L-arginine on blood lipids is probably contribute to the catabolism of bile acids by Lactobacillus plantarum producing more amino acid metabolites. However, more research is needed to explore the underlying mechanisms of what components of the symbiotic increase Glu-Phe and L-arginine. The actions of these altered plasma metabolites may represent an avenue for the prevention and treatment of HFD-induced obesity via symbiotic supplementation.

Although our symbiotic formula did not deliver a powerful synergistic impact on weight loss, it effectively balanced the beneficial properties of each component, but further research is required to explain the underlying mechanisms. In general, most previous experimental research on symbiotics in animals showed dose-dependent effects, as did this study. The low dosage of our symbiotic only partly improved the metabolic disorders induced by obesity in rats, and the potential long-term impacts have not been demonstrated. We propose that a symbiotic may require specific interventional doses to achieve significant biological effects. Further, these types of supplements may be more appropriate for use on a changed diet and lifestyle basis. 

## Figures and Tables

**Figure 1 nutrients-15-00956-f001:**
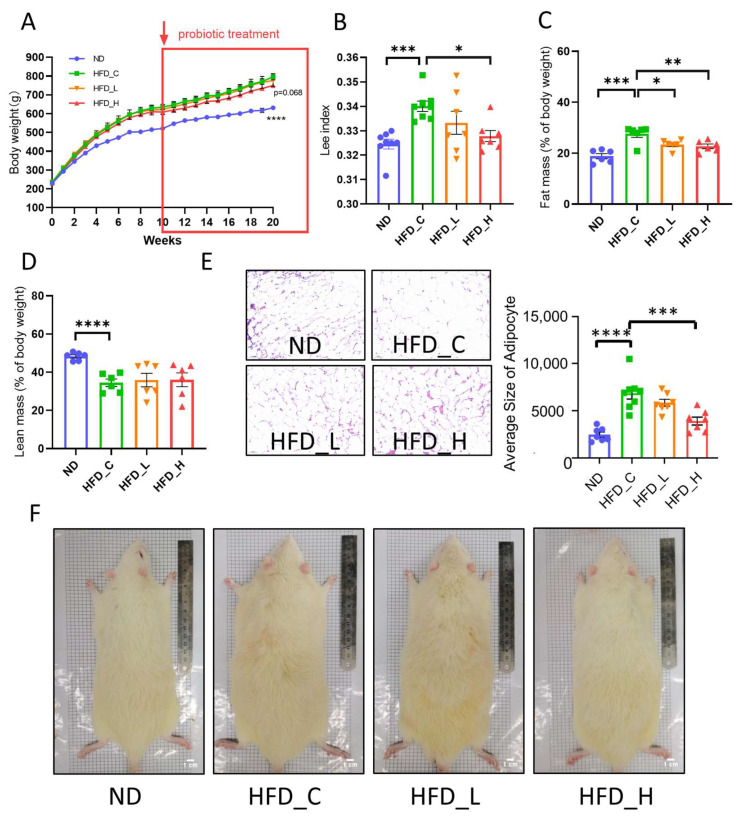
Effects of symbiotic supplementation on body weight and body fat in rats. Comparative measurements of (**A**) body weight, (**B**) Lee index, (**C**) fat mass and (**D**) lean mass in the four treatment groups. (**E**) Typical hematoxylin and eosin (H&E) staining of epididymal fat from rats and average epididymal adipocyte size in the four groups. (**F**) Representative photos of rats from each group. Data are presented as mean ±SEM; * *p* < 0.05, ** *p* < 0.01, *** *p* < 0.001 and **** *p* < 0.0001 compared with the HFD_C group. ND = normal diet group; HFD_C/L/H = high-fat-diet control/high-fat-diet + low-dose symbiotic/high-fat-diet + high-dose symbiotic groups.

**Figure 2 nutrients-15-00956-f002:**
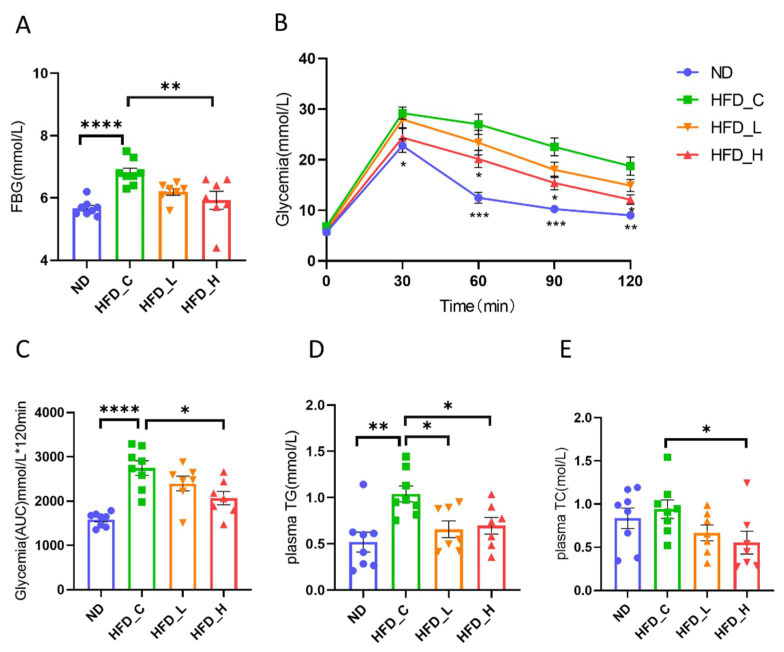
Effects of symbiotic supplementation on metabiotic parameters in rats. Comparative measurements of (**A**) fasting blood glucose (FBG), (**B**) intraperitoneal glucose tolerance test (IPGTT), (**C**) IPGTT area under the curve (AUC), (**D**) triglyceride (TG) and (**E**) total cholesterol (TC). Data are presented as mean ±SEM; * *p* < 0.05, ** *p* < 0.01 and **** *p* < 0.0001 compared with the HFD_C group.

**Figure 3 nutrients-15-00956-f003:**
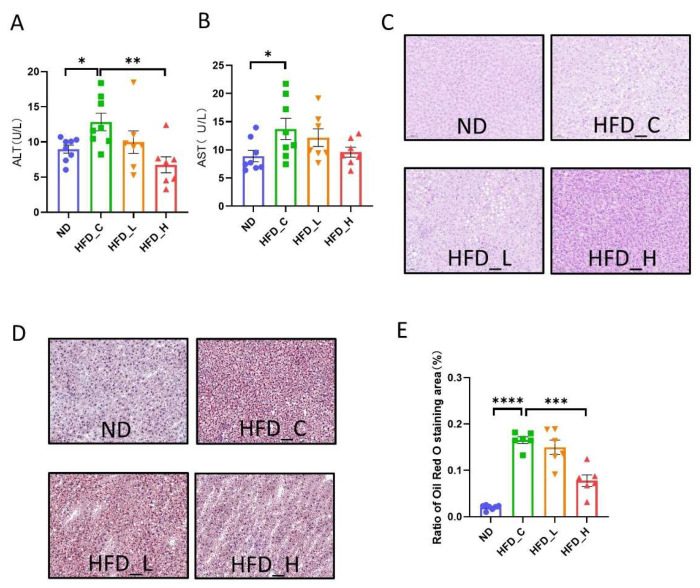
Effects of symbiotic supplementation on fat accumulation and enzyme activity in the liver in rats. Comparative measurements of (**A**) alanine aminotransferase (ALT) and (**B**) aspartate aminotransferase (AST) in the four groups. Representative staining of livers from the four groups with (**C**) H&E and (**D**) oil red O. (**E**) Ratio of oil red O staining area (%). Data are expressed as mean ± SEM; * *p* < 0.05, ** *p* < 0.01, *** *p* < 0.001 and **** *p* < 0.0001 compared with the HFD_C group.

**Figure 4 nutrients-15-00956-f004:**
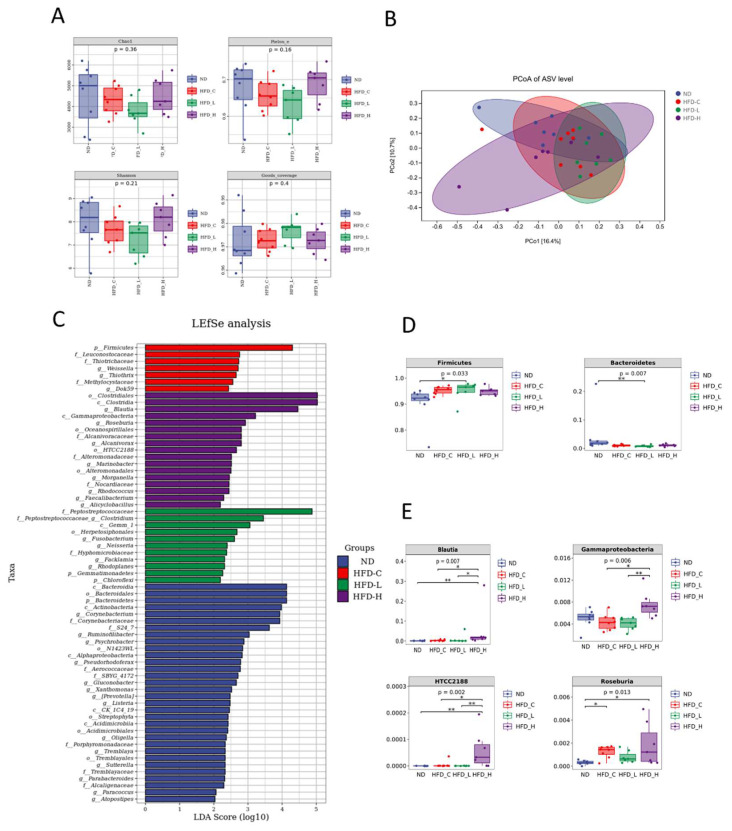
Supplementation of symbiotics altered the gut microbiota composition. (**A**) Alpha diversity, (**B**) beta diversity, (**C**) linear discriminant analysis effect size (LEfSe) histogram showing microbiota significantly enriched within each group. (**D**) The phylum-level proportional abundances of Firmicutes and Bacteroidetes. (**E**) The abundance of core microbiota significantly altered in HFD_H compared with HFD_C. * *p* < 0.05 and ** *p* < 0.01.

**Figure 5 nutrients-15-00956-f005:**
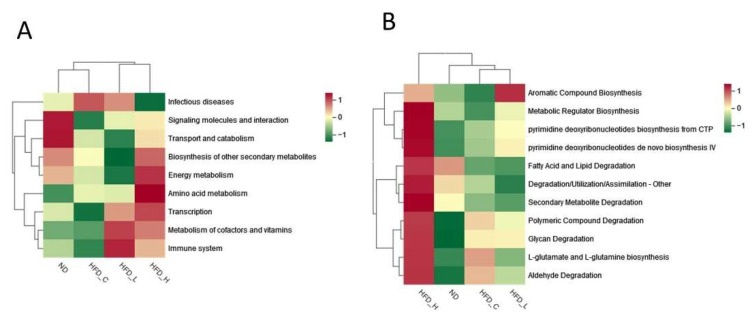
Functional relative abundance of intestinal flora based on the Kyoto Encyclopedia of Genes and Genomes (KEGG) and Metabolic Pathways from All Domains of Life (MetaCyc) databases. (**A**) Heatmap of metabolic function clustering of KEGG. (**B**) Heatmap of metabolic function clustering of MetaCyc.

**Figure 6 nutrients-15-00956-f006:**
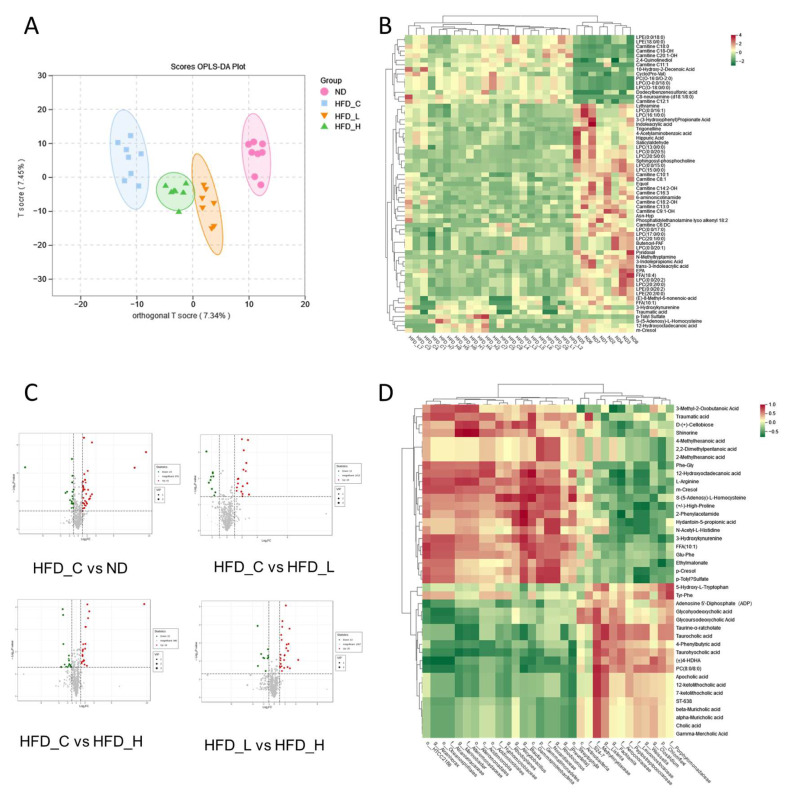
Supplementation of symbiotics altered the metabolic patterns. (**A**) Orthogonal partial least squares-discriminant analysis (OPLS-DA) score plots of plasma metabolic profiling of the ND, HFD_C, HFD_L and HFD_H groups. (**B**) The relative amount of metabolites concurrently varied in the ND, HFD_C, HFD_L and HFD_H groups is transformed into Z scores in the heatmap. (**C**) Volcano plots for differentially expressed metabolites in two-group comparisons. (**D**) The relationship between the 46 different metabolites and the 67 core microbiota in HFD_C and HFD_H is estimated by Spearman’s correlation analysis.

## Data Availability

The data presented in this study are available on request from the corresponding author.

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
