# Peer review of "A Novel Symbiotic Formulation Reduces Obesity and Concomitant Metabolic Syndrome in Rats by Raising the Relative Abundance of Blautia"

_nutrients, 2023, doi:10.3390/nu15040956_

Round 1

Reviewer 1 Report

This is a well written manuscript discussing the authors’ findings using a novel symbiotic formulation reduces obesity and concomitant metabolic syndrome in rats. The authors chose 

 two Bifidobacerium species, three Lactobacilli species, and four prebiotics to make a new synbiotic. High- or low-dose of synbiotic were administered to the rats and recorded biochemical indicators to assess the biological effects in a high-fat-diet-induce rat model. The authors successfully showed that high-dose synbiotic supplementation was effective in reducing obesity and concomitant metabolic syndrome. High-dose synbiotic also significantly increased the abundance of Blautia, which was negatively correlated with taurocholic acid and the main differential metabolites involved in amino acid and bile acid metabolism. It is very interesting that this was shown in a dose dependent fashion. 

There is scant data to show that each separate probiotics and prebiotics impact obesity. The authors combine five probi-119 otic strains (B. animalis ssp. lactis 420, B. breve B-3, L. acidophilus JYLA-191, L. rhamnosus 120 JYLR-005 and L. plantarum JYLP-002) mixed with four prebiotics (apple pectin, inulin, 121 fructo-oligosaccharides and cranberry powder). The formulation of the combinations is controlled well, but the reader cannot deduce from the data which products are truly most responsible for the changes seen in HFD-L mice vs. the HFD-H mice. I appreciate that fecal and serum samples were utilized to evaluate LC-MS/MS of the metabolites. 

I find it surprising that symbiotic treatment reduced body weight and body fat in obese mice more than the non-obese mice. I would anticipate if the product is truly effective then it would reduce weight in all the mice, but with different impact in the obese vs. non-obese mice. This is not well explained in the discussion and I would like for the authors to offer more in the discussion about why the think these results happened. Figure 2 D is fascinating and makes sense that there would be improvement in dysregulated glycemic function and lipid function in the obese mice vs. the non-obese mice. 

I recommend that the authors utilize a more stringent statistical analysis for the high throughput data evaluated with LC-MS/MS. T-tests  and two way ANOVA are acceptable for normally distributed data, but it is highly unlikely that such large datasets evaluating metabolites is normally distributed. 

I do think it is quite a stretch to say that these changes represent “positive therapeutic effects on obesity” and this should be re-worded to be more quantitatively representative of the results shown in their data figures. 

The word “insulin” is misspelled multiple times throughout the manuscript. Line 57 is the first time it is noted. The authors need to carefully proofread this and correct before this can be published. 

This is an interesting study and I agree with the authors that more data is required to further study the underlying mechanisms and that this is novel work. I would recommend publication after the discussion and statistical considerations I have offered are addressed. 

Author Response

Dear reviewer:

We have received your comments, please see the attachment for the response.

Reviewer 2 Report

The manuscript “A novel symbiotic formulation reduces obesity and concomitant metabolic syndrome in rats by raising the relative abundance of Blautia” is well written and the experimental design well thinking but could be improved, in my opinion. I have three main concerns with this study.

One of my concerns is that the study considers cranberry powder a prebiotic and does not sustain this information in any study. I have a seriously doubt how a cranberry powder can be a prebiotic, because one of the requirements for being a prebiotic is that it should be resistant to gastrointestinal digestion (GIT). And I have a huge doubt that the polyphenols there, will be intact through GIT, particularly because they were not encapsulated, unless authors cite a study where it was proven the prebiotic properties of cranberry powder. Authors may also overlap this concern by calling the mix a symbiotic supplemented with a rich source of polyphenols.

My main concern is that I couldn’t understand throughout the manuscript why authors decided to join these specific components, they described each beneficials, but they do not explain why they think this mix could be better than other one. Authors do not create a clear hypothesis, and in the discussion, they never make a relation between the results observed and each component. It seems all study is only to “cook and look”, which in my opinion decreases the novelty and the scientific soundness of the study.

The last concern is related to the significance of content and the target audience of the manuscript. I can understand that is necessary to see the effect on the causing effect (HFD) and I agree, but I do not agree that authors idealize a supplement for people to eat whatever they want, which I think will raise some ethical concerns. I’m not expecting that an obese will start a treatment with a symbiotic with any other changes in the diet, because there are no miracles, and in my opinion this study should progress to the effect of the symbiotic in an obese which starts a normal diet. It should be quiet interesting compare ND vs HFD vs HFD-ND with addition and no addition of the symbiotic at different doses. For the group HFD-ND I would start giving a ND to the control rats (HFD-C) after the 20 weeks, with no addition and addition of the same symbiotic doses and compare with the results here presented. This way authors may prove the symbiotic impact and their dose dependance on HFD groups, but also an expected satisfactory result when accompanied by a ND to obese people.

Abstract

·  Line 22. Synbiotic or symbiotic?

·  Line 23. Bifidobacterium

· Line 25. High- or low-dose of symbiotic were administered to the rats and biochemical indicators recorded to assess the biological effects in a high-fat-diet-induce rat model.

·  Line 26. mechanisms

Keywords

Authors should not repeat words from the title. Remove obesity and symbiotic and add keywords such as overweight, prebiotics, probiotics and polyphenols or inulin, FOS, pectin, cranberry powder, Bifidobacterium, Lactobacillis

Introduction

·   Line 49. it was interesting to know a little bit more about this imbalance

·   Line 53. …when fed with Bifidobacterium animalis ssp.lactis….

·  Lines 58-66. Authors start to write about inulin followed by pectin and then oligofructose. In my opinion the order should be inulin, oligofructose and pectin, since the oligofructose authors are describing are fructo-oligosaccharides (FOS) obtained from the polysaccharide inulin and Pectin is a completely different story.

·  Line 66-71. paragraph out of place or not well written and introduced. Why authors include here the polyphenols, these should be included in a new paragraph after lines 71-73. Polyphenols are not dietary fibers, what authors may argue is that some dietary fibers depending on their origin may have attached polyphenols and then being an added value to the fibers, but it seems is not the case.

·   Line 75. again, synbiotic or symbiotic? Please check all the manuscript and use only one

· Line 96. where is it stated that cranberry powder is consider a prebiotic? There’s no reference for that and I have my serious doubts

·   Line 99. what is Blautia? And what is its importance? Is the first time that Blautia appears, readers not in the field will ask “what is this?”, “What’s the purpose?”

Material and methods

Line 118. section 2.2, there are no encapsulation the symbiotic is a simple mix of probiotics, prebiotics, and cranberry powder, if though the components of cranberry powder cannot reach the intestinal part intact.

Results

· When authors write lower than, higher than, how much? please quantify in number x-fold or percentage. Authors may use brackets.

· Figure 2A. p=0.06 in the figure it is not relevant, the sentence on line 233 is enough.

· Lines 262-264. but not enough to reach a similar situation of a ND, it should be interesting here to see if a group starting a ND after a HFD, could reach a more close position of a ND group, you may use the control group and start a ND with no addition and addition of increasing doses of the symbiotic.

· Lines 268-281. Not everyone is used to this type of graphs, authors should explain better what Figure 4E means, because is difficult to understand why after showing increasing of the same gut microbiota (present in all the groups) in the end appears a graph that shows that the groups are composed of a completely different gut microbiota. Authors need to be more precise.

· Figure 4B. Can authors please discuss why this PCo1 which explains only 16.4% of the data and PCo2 only 10.7% and even though is presented and discussed on line 272 “…but it did impact beta diversity (Figure 4B).” Also, the discussion presented seems to me very redundant, please be more detailed on the presentation of this result explaining what you are seeing in the PCoA.

· Figure 4. I would change this figure into three, as it is, the graphs are too small and is difficult to understand even when zooming.

·  Line 291. repeating word

· Figure 6. It is very difficult to understand figure labels, particularly in Figure 6C

·  Figure 6A. please explain the low T scores percentage

·Line 309. I would like to have an overview on the metabolite differences observed which makes the groups distinction and then in the discussion part related these with the components of the symbiotic

· Line 339. Figures and tables should not be a section, but each one being close and after being referred in the text. Is annoying for the readers going up and down to see the figures and read the correspondent section.

· Lines 381-382. I think this information should be on materials and methods section

Discussion

·      Line 386. Shouldn’t be affected and not effected?

·      Lines 412-413. And why authors think this happen?

·     Line 429-430. “This extraordinary result encapsulates….” Not the best sentence

·   Line 462. I would like authors explain the sentence “…it effectively balanced the beneficial properties of each component….” And specify which results show this effectively balanced for each component…

Author Response

Dear reviewer

We have received your comments, please see the attachment for the response. Thank you.

Reviewer 3 Report

Dear Authors,

Thank you for submitting this interesting paper investigating the effect of high- or low-dose of synbiotic administered to the rats and their biological effects in a high-fat-diet-induce ra. Overall I had a great pleasure reading this manuscript, as it presents a novel point of view and it is carefully prepared.

Hovewer, please consider the following points: 

  1. L123 – the authors wrote that the exact formula of the synbiotic is listed in Table 1. Hovewer there is no Table provided. Please add it to the manuscript.
  2. L121 – there is no information regarding the prebiotic part of the novel synbiotic that was used by the authors. How much of ech component was given in high- and low-dose of synbiotic given to rats.
  3. L121 – regarding the prebiotic, there is also no data concerning whather the authors used a manufacturers products, if yes some information on that should be given.
  4. L121 -  concerning the inulin, used as a part of synbiotic, there is no information concerning the DP of inulin and from which natural source it was extracted. Please add these data.
  5. I am curious wheter the authors have measured some plasma and liver parametres reflecting the antioxidant capacity. If yes, would it be possible to incorporate it and discusse it?
  6. I also suggest to carefully read the paper once again as some misspelings occurs.

Author Response

(The authors gave the same response as above.)
